# War-related trauma in narratives of Gazans: challenges, difficulties and survival coping

Bilal Hamamra[1], Fayez Mahamid[1] , Dana Bdier[1] and Mai Atiya[2]

[1]Faculty of Humanities and Educational Sciences, An-Najah National University, Nablus, Palestine and [2]Psychological and Educational Counseling Department, Al-Quds Open University, Gaza Branch, Palestine

## Research Article

war-related trauma; collective trauma; survival coping; Gaza Strip; Palestine

**Corresponding author:**
Fayez Mahamid;
Email: mahamid@najah.edu

## Abstract

The Israeli escalation of violence and oppression in Gaza, particularly following the events of October 7, 2023, has deepened the trauma and exacerbated the already grievous conditions of dispossession and exploitation faced by Palestinians. The present exploratory work sought to analyze war-related traumatic experiences among Gazans following the recent Israeli genocide against Gazans. Thirty participants (mean age for males = 32.45 years, SD = 10.13; mean age for females = 30.28 years, SD = 9.15; range 19–57) were recruited via snowball sampling. Interviews were analyzed through thematic content analysis. Thematic content analysis of the interview transcripts led to the identification of five main themes: (1) challenges and difficulties faced Gazans, (2) traumatic experiences related to war, (3) feelings and emotions among Gazans living in refugee camps, (4) how do Gazans perceive the future and (5) survival coping employed by Gazans following the on-going genocide against the Gaza Strip. The findings of this study highlight the profound humanitarian crisis in Gaza, calling for urgent attention to the ongoing crisis in Gaza and advocating for comprehensive humanitarian support and psychological interventions to address the deep-seated trauma and help rebuild lives shattered by conflict.

## Impact statement

Since the events of October 7, 2023, Gazans have been subjected to a series of devastating and traumatic experiences. These events included the large-scale destruction of infrastructure across the Gaza Strip, the significant loss of life during the war and the displacement of hundreds of thousands of Gazans, many of whom sought refuge in the Rafah area. These traumatic experiences have had a profoundly negative impact on the mental health of the people in Gaza. The results of the current study shed light on the ongoing psychological trauma faced by Gazans, emphasizing the severity of their situation. It also explores the coping strategies they employ to manage the distressing events they have endured. Additionally, the findings highlight the urgent need for immediate mental health interventions aimed at addressing the psychological trauma experienced by individuals in Gaza. Such interventions should focus on alleviating trauma and promoting positive coping strategies to enhance the mental resilience of those affected by the political conflict.

## Introduction

On October 8, 2023, the Israeli occupation launched Swords of Iron in response to the Al-Aqsa Flood on October 7, 2023, the first-ever coordinated incursion into the Gaza envelope by Hamas, Palestinian Islamic Jihad and other Palestinian armed groups. This conflict escalated into a war characterized by the Israeli genocidal assault against Gazans. While the Gaza Strip bears one of the Israeli deadliest aggressions, the West Bank is facing various forms of oppression and collective punishment aimed at intimidating Palestinians (Veronese et al., 2024).

The trauma endured by the Palestinians is a result of the Israeli systemic strategies of domination, colonial dispossession and ethnic cleansing (Halper, 2015; Pappé, 2015). This Israeli regime of territorial and bureaucratic control spans all dimensions of life in Palestine, from airspace to subterranean spaces. The Israeli deliberate inflictions of violence, restrictions on movement, economic blockades and destruction of homes and infrastructure constitute acts that erode the will and coherence of Palestinian society. Weizman (2017) and Rouhana and Sabbagh-Khoury (2020a) emphasize that these actions are designed to fragment and weaken the Palestinian social fabric, making survival and resilience formidable challenges for Palestinians.

### Genocide-induced trauma

The concept of trauma within the Israeli–Palestinian context can be dissected into three main constellations: collective, colonial and transgenerational trauma. Collective trauma reflects the

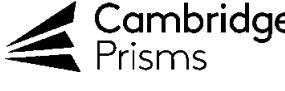



shared experiences of violence and grief that ripple through the community, manifesting in widespread psychological distress and a pervasive sense of loss (Chaitin, 2011; Bar-On, 2016). The communal nature of this trauma means that entire communities experience and respond to the same traumatic events, creating a shared narrative of suffering that is reinforced by repeated episodes of violence. Colonial trauma captures the psychological and social effects of prolonged exposure to colonial suppression and control that strip away Palestinians' rights, resources and autonomy, embedding a state of chronic instability and insecurity (Fanon, 1963; Memmi, 1965).

Colonial trauma disrupts social structures and cultural identities, creating a pervasive sense of powerlessness and existential threat. Transgenerational trauma extends the discourse to the enduring impacts that these experiences have on successive generations. The cyclical nature of trauma ensures that the psychological scars and sociocultural disruptions are inherited by following generations. This perpetuates a legacy of suffering and resistance within families and the wider community (Dekel and Goldblatt, 2018; Qouta et al., 2019). Children grow up in an environment marked by the traumas of their parents and grandparents, leading to a continuous transmission of trauma that shapes their psychological and social developments. These forms of trauma intersect and compound each other, creating a web of psychological, social and cultural challenges that Palestinians must traverse daily. The Gazacide and its associated traumas disrupt individual lives and threaten Palestinian society, making resilience and coping mechanisms crucial for survival and continuity.

The collective trauma experienced by Palestinians, particularly in Gaza, is influenced by the Israeli systematic deprivation of fundamental human rights and essential facets of quality of life. This includes deprivation across environmental, relational, psychological, economic and physical domains (Hamamra et al. 2024). Such deprivation subjects Palestinians to traumatic burdens that surpass the typical responses to specific traumatic events (Finkelstein, 2000).

Further exacerbating this collective trauma are the pervasive Israeli military surveillance and control mechanisms imposed on Gaza. The use of aerial drones, stringent permit regulations for workers and the medically ill and severe restrictions on the entry of basic goods into the Strip systematically disrupt the daily lives of Palestinians. These measures not only fracture the sense of continuity and coherence within the population but also reflect a deeper legacy of social and psychological subjugation (Falah, 2005).

The stringent permit regulations impede access to essential services and economic opportunities, further entrenching poverty and dependency. Meanwhile, the severe restrictions on goods exacerbate humanitarian crises, limiting access to food, medical supplies and other necessities, thus perpetuating a cycle of suffering and deprivation. This relentless surveillance contributes to a pervasive sense of hopelessness and despair. Such trauma not only impacts the current generation but also leaves an indelible mark on future generations. The social and psychological subjugation experienced by Palestinians in Gaza is emblematic of the broader colonial strategies aimed at undermining their resilience and eroding their cultural and social cohesions (Mahamid et al., 2024).

### Educide and medicide

Gazacide has deepened the trauma and exacerbated the already grievous conditions of dispossession and exploitation faced by the Palestinian population. This intensified phase of conflict saw the implementation of policies that actively promoted expulsion and displacement, severely undermining the educational infrastructure through acts of "schoolcide" and epistemicide – the systematic destruction aimed at erasing the educational legacy of the community. Such actions reinvigorated the historical wounds associated with the Nakba, the catastrophic displacement and murder of Palestinians in 1948 (Pappé, 2015).

The Israeli attacks on Gaza's educational sector have been meticulously documented. The Office for the Coordination of Humanitarian Affairs reported in 2023 the specific targeting of academics. According to the Euro-Med Monitor, the numbers are stark: 17 individuals holding professorial titles, 59 with doctoral degrees and 18 with master's degrees were among those targeted. These figures, however, are likely underestimations due to significant challenges in documentation. These challenges stem from movement restrictions, disrupted communications and widespread internet shutdowns. Their loss represents not only a direct attack on individuals but also a profound blow to the intellectual and cultural capital of Gaza.

Furthermore, the systematic targeting of Gaza's educational infrastructure, such as the destruction of Islamic University, Al-Azhar University and the complete razing of Al-Israa University after its repurposing as military barracks, as captured in a video released by Israeli media (Euro-Mediterranean Human Rights Monitor, 2024), illustrates a blatant disregard for the principles of international humanitarian law. These actions not only cripple the immediate educational environment but also aim to dismantle the cultural and intellectual fabric of Palestinian society (Mahamid et al., 2024). The destruction of these universities symbolizes an attack on the aspirations and future prospects of an entire generation. Educational institutions are pillars of hope and progress, and their destruction leaves a void that is difficult to fill. The loss of these academic centers disrupts the transmission of knowledge, hinders research and development and limits opportunities for students to pursue higher education and career advancement. Moreover, the compounded effects of these actions signify a broader, more insidious form of cultural and intellectual genocide. By targeting educational institutions, the strategy aims to erase the historical and cultural identity of the Palestinian people, making it increasingly difficult for the community to recover and rebuild (Bdier et al., 2023).

This highlights the urgent need for international recognition and intervention to protect the educational and cultural heritage of the Palestinian people (Bdier et al., 2023). International intervention is crucial to not only halt the immediate destruction but also rebuild and protect the educational foundations necessary for the future of Gaza's youth. Without such support, the long-term effects of these targeted actions could lead to a significant loss of cultural identity and educational progress, further entrenching the cycle of poverty and conflict. Thus, it is imperative for the global community to recognize and address these acts of cultural genocide, ensuring that the rights and heritage of the Palestinian people are preserved and respected (Mahamid et al., 2023).

Medical facilities, already under immense stress, face collapse during power outages, critically impairing life-saving services and leading to increased fatalities, particularly among newborns and critically ill patients (UNICEF, 2023). Jeffrey Alexander's notion of collective trauma is vividly illuminated here, where a community experiences a horrendous event that leaves indelible marks upon their group consciousness, altering their memories and future identity in fundamental and irrevocable ways (2004). This trauma

fosters long-term mental health crises and perpetuates a cycle of trauma across generations.

The current plight of an entire population and the imposition of colonial, collective and transgenerational psychological burdens further exacerbate the risk of genocide, as defined by causing serious bodily or mental harm to members of the group (Finkelstein, 2000; Shalhoub-Kevorkian, 2014; Jabr and Berger, 2017; Veronese et al., 2023). These strategic and multifaceted impacts – ranging from the immediate deprivation of life-sustaining services to the long-term psychological and cultural destruction – reflect a disregard for international humanitarian principles.

The systematic approach to undermining Palestinians highlights an intent to destroy the social and cultural existence of this community. This scenario exemplifies collective trauma, reshaping their collective memory and identity in ways that will likely influence Gaza's social dynamics and community relationships for generations to come. The cycle of trauma perpetuates a legacy of suffering, shaping the collective psyche and behavior patterns across generations. The continuous exposure to violence, deprivation and psychological warfare leads to long-term mental health crises, chronic stress and a pervasive sense of insecurity and hopelessness. They result in the erosion of community cohesion, the fracturing of social structures and the dismantling of cultural identity (Mahamid and Berte, 2020).

The systematic dehumanization of Palestinians is closely linked to the notion of collective trauma (Jabr and Berger, 2017). Alexander's concept of collective trauma is particularly relevant here, noting that such trauma occurs when community members are profoundly impacted by horrific events, leaving indelible marks on their group consciousness. This reshapes their collective memory and fundamentally alters their future identity (Alexander, 2004). The Israeli narrative, which diminishes Palestinians to subhuman status, combined with the aggressive retaliatory measures following October 7, 2023, embeds a persistent narrative of persecution and suffering within the collective psyche of the Palestinian people. Furthermore, Gregory Stanton's genocide framework, especially the stages of classification, symbolization, dehumanization and discrimination, vividly characterizes the actions of the Israeli state against Palestinians. These stages lay the groundwork for genocide by systematically dehumanizing the target group and justifying the escalation of violence against them. This process not only reinforces the traumatic experiences of the Palestinians but also embeds these experiences as central elements of their collective identity, perpetuating a cycle of suffering and resistance within the community (Stanton, 2017).

The persistent portrayal of Palestinians in dehumanized terms by Israeli figures, such as Defense Minister Yoav Gallant who referred to Gazans as "human animals" (Ben-Naftali et al., 2024), legitimizes ongoing aggression and cements these traumatic experiences within the group's identity. This dehumanization influences their self-perception and their view of their aggressors, further entrenching the cycle of trauma. Previous studies on trauma in Palestine have focused on specific aspects of the population's vulnerability to the atrocities committed by the Israeli occupation, while others have examined the resilience of Palestinians in the face of such adversity. For instance, Marshall (2014) offered a critical perspective on the collective trauma experienced by Palestinian children. They underscore the importance of incorporating sociopolitical realities into the understanding and treatment of trauma, highlighting how these realities shape the experiences and responses of young Palestinians. Altawil et al. (2008) further elucidated the psychological toll on children in Gaza, noting a significant prevalence of post-traumatic stress disorder resulting from continuous exposure to violence. Their research reveals the profound impact of ongoing conflict on the mental health of children, emphasizing the need for targeted interventions to address the specific trauma experienced by this vulnerable group. Moreover, Afana et al. (2020) explored culturally embedded coping mechanisms within the Palestinian community. Their work highlights the resilience of Palestinians, demonstrating how cultural practices and community support systems provide crucial mechanisms for managing and overcoming trauma. However, this resilience is currently being tested under extreme conditions, as the ongoing conflict and escalating violence challenge the community's coping capacities.

### The setting

Within the context of the ongoing genocide in Gaza, one should be mindful to the fact that trauma, collective trauma, extends beyond those who are affected by it. In other words, the effect of the ongoing war on Gaza elicits the sympathy of global witnesses who are traumatized by the Israeli brutal actions, actions that nurture solidarity with Palestinians. For example, Reinhart (2024) articulates a compelling analysis of the global solidarity observed in response to the genocide in Gaza, suggesting a profound psychological and political transformation far beyond the confines of Palestine. As a political anthropologist, he interprets the widespread empathy for Palestinians not as mere sympathy but as a deep, visceral connection rooted in shared histories of intergenerational suffering under Western imperialism and racism. This solidarity, particularly pronounced in the Global South, reflects a collective psychic resonance that transcends geographical boundaries, catalyzed by the relentless visibility of Gaza's plight on social media. Reinhart posits that this global reaction is not simply a passive expression of support but signifies a burgeoning revolutionary subjectivity, a collective ethical awakening that challenges the Euro-American-dominated international order. This paradigm shift suggests a move toward a more interconnected understanding of justice and ethical responsibility in international relations, driven by a shared recognition of historical and ongoing oppression. With respect to Palestinians in diaspora, Abla Abdelhadi (2024) examined the profound psychological trauma experienced by Palestinians witnessing the ongoing genocide in Gaza from afar. She reveals how the constant exposure to the atrocities committed against her people induces severe anxiety, survivor's guilt and a debilitating inability to focus. Abdelhadi's experience underscores the concept of "psychological terrorism," where the indirect exposure to violence triggers intense emotional distress and mental health breakdowns even in those not physically present in the conflict zone. In response to these overwhelming challenges, displaced Gazans have developed various coping strategies characterized by strong communal bonds, faith and adaptive resilience. Social support networks – including family, friends and newly formed connections within displacement camps – provided essential emotional and practical assistance. Faith served as a crucial source of hope and meaning amidst the on-going genocide against Palestinians in Gaza. Additionally, humor and emotional expression were utilized by some individuals to navigate the severe hardships they faced. Given that earlier research (Marshalln and Sousa, 2017; Mahamid et al., 2023, 2024) showed Palestinians in

the Gaza Strip endure various types of war-related, political and collective traumas due to the Israeli war on the Gaza Strip, our study aimed to explore the following questions:

*First: How do Palestinians living in the Gaza Strip narrate war-related trauma experiences after the Israeli war on the Gaza Strip?*

*Second: How do Gazans perceive the future following the Israeli war on the Gaza Strip?*

*Third: What are the predominant emotions experienced by Gazans following the events of October 7, and Israeli war on the Gaza Strip?*

*Fourth: What are the survival coping employed by Gazans following the recent war on the Gaza Strip.*

## Methods

### Participants

The study participants were 30 displaced Gazan, selected from internally displaced Palestinian camps in the city of Rafah during the recent conflict in the Gaza Strip. The group comprised 14 females and 16 males, aged between 18 and 58 years (mean age for males = 31.43 years, SD = 11.12; mean age for females = 31.23 years, SD = 10.13). All participants were displaced residing in internally displaced camps in Rafah. They were all sufficiently eligible and spoke Arabic to complete the research tasks.

### Instruments and procedures

The qualitative data were collected through 30 semi-structured interviews with Palestinian displaced living in internally displaced camps in Rafah. All participants (both interviewees and the interviewer) were native Arabic speakers. Research assistants in each camp acted as gatekeepers in recruiting participants. The data collection process began with interviewing the research assistants to explain the aims and purposes of the study. The second step involved informing them about the research activities, the total number of participants to be interviewed and the "snowballing" technique used for selecting participants from those who accepted the invitation. The survey questions used in the interviews were designed to avoid emotional distress. Participants were informed that they could discontinue their participation at any time if they felt distressed. The investigators, licensed mental health professionals, were available for any participant who experienced an immediate negative response to the survey questions. Furthermore, all participants were given contact information for mental health services in case symptoms appeared after the survey. The data were collected during the war on Gaza, with interviews conducted directly in the shelter schools in Rafah. Given the extreme and rapidly changing conditions, research assistants – who were themselves displaced – worked under immense logistical, security and psychological pressures. The continuous bombardment and forced displacement disrupted interview schedules, requiring flexibility in data collection. Interviews were often relocated to different shelters or postponed due to airstrikes, while communication breakdowns caused by power outages and internet blackouts further complicated coordination.

The research assistants faced a number of challenges during data collection, including the risk of being targeted by airstrikes, the lack of logistical resources and the ongoing displacement, which posed a significant obstacle to data collection. The rapid displacement in Gaza made it difficult to reach participants. Additionally, the difficult

security conditions and the reluctance of some women to participate due to their psychological distress and inability to talk about what they experienced during the war were also significant challenges.

To mitigate these difficulties, research assistants employed adaptive strategies, such as relying on word of mouth to identify potential participants, conducting interviews in the safest available locations and adjusting the interview format based on participants' emotional states. Despite the traumatic conditions, the research team remained committed to ethical considerations, ensuring confidentiality, informed consent and emotional support for both participants and researchers.

The study received approval from the An-Najah Institutional Review Board (IRB) before data collection began. The interviews aimed to gather information about war-related trauma, difficulties, challenges and coping strategies among Palestinian displaced living in internally displaced camps in Rafah during the recent Gaza conflict. Interviewees were provided with an information sheet detailing the research agenda. The shortest interview lasted approximately 35 minutes, while the longest lasted 60 minutes; most interviews were around 50 minutes.

### Data analysis

All interviews were audio recorded and transcribed into Arabic by a native-speaking researcher. The written transcripts were analyzed using thematic content analysis (TCA) methodology (Parker, 2004) to identify the main themes emerging from the material. A bottom-up, data-driven text analysis approach was applied to extract categories from the raw data (Strauss and Corbin, 1990). Each interview was carefully examined to identify concepts and statements containing similar words. The analysis process included the following steps: (a) conducting open coding analysis to derive the main themes from the participants' narratives, (b) coding and organizing the themes into structured texts and (c) discussing and reaching agreement on categories or subcodes with five judges.

### Coding reliability

A reliability test of the coding was conducted, achieving 92% consistency with the author's original coding (Cohen's kappa = 0.94). Cohen's kappa is a statistical measure used to assess inter-coder reliability, with a coefficient of 0.80 or higher considered acceptable.

### Results

TCA of the interview transcripts led to the identification of five main themes: (1) challenges and difficulties faced Gazan after the war on the Gaza Strip, (2) traumatic experiences related to war on the Gaza Strip, (3) feelings and emotions among Gazans living in the internally displaced camps, (4) how do Gazans perceive the future and (5) survival coping employed by Gazans following the recent war on the Gaza Strip.

### Theme 1: challenges and difficulties faced Gazan after the war on the Gaza Strip

The majority of participants in current studies have pointed out many challenges and difficulties they encountered after the war on the Gaza Strip. These challenges included the destruction of their homes, displacement from northern Gaza to the Rafah area and living in a refugee camp that lacks basic necessities. Additionally,

many lost several family members during the Israeli invasion of Gaza and faced many harsh and painful conditions. Indeed, one participant explained:

> We face difficulties in every aspect of life, including basic necessities like water, food, and medicine. Many hospitals in the area have ceased operations due to the Israeli blockade, and even breathing is challenging because it is polluted. In the refugee camps, we lack any essential means for survival. (39-year-old female; Jabalia Camp)

Respondents described difficulties and challenges faced by Gazan after the last war on the Gaza Strip:

> Everything is difficult in Gaza. The country is completely destroyed; there's no way we can live here anymore. I haven't been able to see my wife and children for months. I'm here in the camp, and they're less than an hour away from me. Gaza is ruined; look where it was and where it is now. (45-year-old male; Gaza City)

Another participant reported:

> I can't sleep because I'm constantly thinking about my children and grandchildren and how our lives have been destroyed. Look at how we are living in the camp; our lives are very difficult. We have no electricity, no privacy, and we can't sleep comfortably. (41-year-old female; Khan Yunis Camp)

The participants discussed the severe life difficulties they faced after their displacement experience. They described how their lives have become nearly impossible and extremely challenging, lacking even the most basic human necessities:

> Displacement itself is a problem that no mind can bear. I used to live in a house that fully met my needs comfortably. Suddenly, I became homeless, living in a tent that serves as my bedroom, kitchen, and bathroom. I have no electricity, no gas, no water, and not even the basic necessities of life. (52-year-old female; Bayt Lahiya)

### Theme 2: traumatic experiences related to war on the Gaza Strip

Participants in the current study described numerous traumatic experiences they endured after the war on Gaza, which followed the events of October 7. They spoke about witnessing horrifying scenes such as collapsing buildings, bodies in the streets, bloodshed and the sounds of explosions everywhere.

A 47-year-old female from Jabalia camp reported:

> My neighbors were destroyed in front of me, and the house collapsed on them. I had to carry children with their intestines torn apart in my arms. This experience has stayed in my mind until now; it remains in my thoughts

Another respondent stated:

> The hardest event I went through was losing my niece. Not only did I lose my niece, but I also lost 18 other family members. When I received the news about my niece, I was shocked. I was standing in the hallway when I heard my other niece screaming. People came to tell me to hurry because my niece had been martyred. I began to scream and cry; it was the most difficult thing because her baby was only forty days old. My niece died while she was displaced from northern Gaza; she was shot. (44-year-old female; Al-Maghazi Camp)

Other respondents shared the traumatic experiences they encountered during their displacement from northern Gaza to Rafah city.

> Many of my family members were martyred in the war. We were displaced to Rafah and initially lived in a school before moving to a camp. An Israeli airstrike hit a building next to the school in the middle of the night, and we all ran into the street. My son, who is in

third grade, lost his ability to see from the extreme fear when the building was bombed. When I took him to the doctor, he was examined and found to be physically fine, but his vision loss was due to the overwhelming fear. (26-year-old male; Dayr al-Balah)

Another participant added:

> I lost one of my children during the war, and we also lost our entire source of income, leaving us without any means of livelihood. Our lives changed completely; we went from having homes to being refugees, without any of the basic necessities for living. (39-year-old female; Jabaliya)

### Theme 3: feelings and emotions among Gazans living in the internally displaced camps

All participants in the current study conveyed a variety of emotions regarding the situation in Gaza, such as sadness, pain and regret. They also expressed feelings of anger, fear and deep anxiety about what the future holds.

> I feel intense anger about what is happening in the Gaza Strip, which many see as a form of genocide. The cold-blooded killing of Palestinians, the disregard for international laws, and the global silence, coupled with ambiguous media and political coverage, are viewed as support for the Israel occupation and a continuation of the genocide in the Gaza Strip. (44-year-old male; Bani Suheila)

Another participant reported:

> My emotions are mixed and unusual, including fear, sadness, existential anxiety, fear of death, and horror. I am overwhelmed by the daily scenes of displacement, estrangement, destruction, shattered buildings, and separation from loved ones. (47-year-old male; Abasan al-Kabir)

The deep emotional distress as a result of the war on the Gaza strip perpetrated by the Israeli occupation is highly reported by many of the respondent. A 38-year-old female interviewee from Khan Yunis camp mentioned:

> Fear, exhaustion, and confusion are the dominant emotions I'm experiencing at this time. I am afraid of the bombings and the violence directed at us, exhausted from displacement, the psychological stress, and the regression of our lives to the primitive life.

> One participant explained *I am overwhelmed by negative emotions ranging from fear and anxiety to sadness, with the predominant feeling being anxiety about the future.* (48-year-old male; Al-Nusayrat Camp)

### Theme 4: how do Gazan perceive the future

The majority of participants in the current study expressed a pessimistic perspective about the future, believing that there is no longer a future for themselves or their children after the recent war launched by Israel on the Gaza Strip. The absence of hope for the future is one of the most significant responses expressed by the participants in the current study.

> There is no future in Gaza after the recent events. Since my relatives died, I've been worried about my son, afraid he might die like they did. My son has grown up, and I am very scared for him. Life has ended for us here in Gaza. We want to emigrate; there's no place left for us here in Gaza. (49-year-old female; Abasan al-Saghir)

One participant described:

> There is no future left in Gaza. Honestly, I don't know where we will go, but the most important thing is that I don't want to lose any of

my children. I'm constantly worried about the future and scared for them. (36-year-old male; Al-Shaykh Radwan)

Another participant explained:

The future looks very bleak. I won't say it's uncertain because the outlines of this dark future are clear. After the war, a new internal conflict will begin. What I see today is unprecedented security chaos and family problems that don't appear in the media but are very real. The country is full of issues, and family vendettas are resurfacing throughout Gaza. You can hardly pass through any area or street without a minor issue escalating into a major conflict, and this is happening every day. (39-year-old female; Tall Al-Turmus)

### Theme 5: survival coping employed by Gazans following the recent war on the Gaza Strip

Participants in the current study shared a range of strategies that enabled them to endure and stay resilient despite the challenging circumstances they faced during the Israeli invasion of the Gaza Strip and their displacement to the Rafah area. These strategies included positive religious coping and social support. One participant clarified:

The social support I received from my family and friends helped me through the current crisis. Additionally, the people I met at the camp supported me and became like family. We have been together for more than nine months now. (37-year-old female; Nazla)

A participant stated that "*My faith in God has helped me overcome the hardships we've faced. I trust in God and believe that this crisis will end soon*" (39-year-old male; Barbara).

One participant who displaced from Jabalia camp reported:

What helped me overcome my hardships was holding onto a sense of humor and continuously expressing my feelings honestly, even when they were painful. The support I received from my family was also incredibly helpful, as was my devoted love for my children and family members. (34-year-old female; Jabalia Camp)

### Discussion

The present work explored war-related trauma, challenges, difficulties and survival coping in narratives of displaced Gazans during the recent Israeli war on the Gaza Strip. Thirty participants shared experiences related to war trauma, hope for the future and survival coping they employed after being displaced from various areas of the Gaza strip to the Rafah area following the war on the Gaza strip.

This study, to our knowledge, is the first to specifically focus on the experiences of Palestinians in the Gaza Strip, exploring their personal narratives surrounding war-related trauma, and the difficulties and challenges they face in the context of ongoing genocide. The study offers a comprehensive examination of the survival and coping strategies employed by Palestinians in Gaza, shedding light on how they traverse the psychological, emotional and social tolls of prolonged political violence. The study provides valuable insights into the resilience and agency of Palestinians living under extreme and prolonged traumatic conditions.

In the aftermath of the genocide, Gazans are grappling with challenges that extend far beyond physical displacement. The participants' accounts highlight the severity of the humanitarian crisis, where basic necessities such as water, food and medical care are scarce. The blockade has exacerbated these conditions, crippling healthcare facilities and leaving residents with limited access to

essential services. Previous studies have highlighted the suffering experienced by the people of Gaza following the events of October 7. These studies specifically reveal the negative impact of the most recent war on mental health and well-being, with a particular focus on affected groups in the Gaza Strip (Kienzler et al., 2024; Veronese and Kagee, 2024; Zivot et al., 2024).

The physical destruction caused by the conflict has not only displaced individuals but also severed familial bonds, leading to a profound sense of isolation. The destruction of homes and infrastructure has rendered Gaza uninhabitable for many, creating a reality where even the simplest act of reuniting with loved ones is fraught with difficulty. This isolation deepens the emotional strain, leaving many feeling disconnected and helpless in the face of overwhelming devastation.

The drastic transition from a stable home to makeshift living conditions in tents illustrates the extreme precarity faced by displaced Gazans. The loss of basic amenities and the forced adaptation to a life of discomfort and instability underscore the severity of the displacement. The psychological impact of such a dramatic shift is profound, contributing to a pervasive sense of insecurity and vulnerability (Buheji, 2024).

The psychological and emotional scars left by the violence are evident in the participants' accounts, which reveal the horror of witnessing death and devastation firsthand. The intense psychological toll of such experiences is manifested in persistent mental health challenges, where the memories of violence remain deeply ingrained (Mahamid et al., 2024).

The personal grief experienced by those who have lost multiple family members in the conflict is overwhelming. The sheer magnitude of loss, particularly in such tragic circumstances, exacerbates the emotional distress, leading to a prolonged state of mourning and sorrow. The impact of these losses is not only personal but also communal, as entire families and communities are left to grapple with the aftermath of the violence (Mahamid and Berte, 2020).

The conflict has also had a severe psychological impact on children, with some experiencing trauma so intense that it manifests in physical symptoms. The fear and anxiety induced by the bombings and violence have left lasting effects on young survivors, highlighting the urgent need for psychological support and intervention. The enduring nature of this trauma underscores the long-term challenges faced by those who have lived through such harrowing experiences.

The emotional responses of Gazans living in displacement camps are marked by a complex mix of sadness, anger and anxiety. The participants' words reflect a deep-seated frustration with the ongoing violence and a sense of injustice regarding the global response – or lack thereof. The perception of systemic violence, coupled with inadequate international intervention, fuels feelings of anger and resentment, as many see the situation as a continuation of the conflict rather than a resolution.

The pervasive fear and existential anxiety experienced by those in the camps further illustrate the psychological toll of the conflict. The constant exposure to scenes of destruction, displacement and separation from loved ones has created an overwhelming sense of dread and hopelessness. This emotional distress is compounded by the daily realities of living in precarious conditions, where safety and stability are constantly under threat.

The exhaustion and confusion experienced by displaced individuals highlight the disorienting nature of their circumstances. The relentless psychological stress, coupled with the regression to more primitive living conditions, has left many feeling mentally and physically depleted. The ongoing uncertainty about the future only

exacerbates these emotions, making it difficult for individuals to find a sense of peace or normalcy.

The participants' views of the future are overwhelmingly bleak, reflecting a deep sense of hopelessness about their prospects. The ongoing conflict has led many to see their future as uncertain and filled with despair. The fear for the safety of loved ones and the desire to escape the dire conditions in Gaza are common sentiments, with some expressing a strong desire to emigrate in search of a better life.

There is also a widespread anticipation of continued internal conflict and instability, suggesting that the impact of the war will extend far beyond its immediate destruction. The potential for security chaos and social strife looms large, reflecting deep concerns about the future stability of Gaza. This bleak outlook underscores the pervasive sense of uncertainty and fear that permeates the lives of those living in the region. Despite the severe adversity, Gazans have developed various coping mechanisms to endure the ongoing Israeli genocide against them. Social support, religious faith and maintaining a sense of humor have emerged as key strategies for survival. The role of community and familial bonds is crucial, providing emotional and practical assistance in navigating the challenges of displacement. New social connections formed in the camps offer a sense of solidarity and support, helping individuals cope with the stress and uncertainty of their circumstances. Religious faith provides another significant source of comfort and resilience, offering hope and a belief in eventual deliverance from the crisis. The strength of faith serves as a psychological anchor, helping individuals maintain their mental and emotional stability in the face of ongoing adversity. Humor and emotional expression also play a vital role in coping, allowing individuals to manage stress and maintain a sense of normalcy amidst the chaos. The ability to find joy in small moments and share laughter with others helps to alleviate some of the emotional burden, contributing to overall psychological resilience.

## Limitations

The present study contains several basic limitations that need to be acknowledged and addressed. First, the sample is a convenience sample of defined geographic areas and not a random sample. Second, the study was based on the use of qualitative data collected through the use of semi-structured interviews. Third, the data for this study were gathered during the Israeli war on the Gaza Strip, which could have intensified traumatic symptoms among participants. To validate the findings of this study, it's important to explore war-related trauma among Palestinians through longitudinal research designs.

## Conclusion

This study explored the lived experiences of Gazans following the recent war on the Gaza Strip through in-depth interviews with internally displaced individuals. The TCA revealed five key aspects of their experiences: the overwhelming challenges and difficulties they faced, the traumatic experiences endured, the emotional toll of displacement, perceptions of an uncertain future and the coping mechanisms employed for survival.

Participants described the severe hardships encountered, including the destruction of homes, displacement to refugee camps lacking basic necessities and the crippling of healthcare services due to the Israeli blockade. These conditions exacerbated an already dire humanitarian crisis, leaving many without access to essential services. The trauma from witnessing widespread violence, death and devastation has left lasting psychological scars, with many participants continuing to struggle with persistent mental health challenges. Their narratives reflect a deep emotional burden, marked by anger, fear, sadness and frustration at the perceived global indifference to their plight. The continuation of violence and the lack of meaningful intervention have fueled a pervasive sense of hopelessness and despair.

The future, as perceived by the participants, appears overwhelmingly bleak. Many expressed a desire to emigrate, driven by fear for their own safety and that of their loved ones. Concerns about ongoing instability and the potential for further conflict contribute to a deep sense of uncertainty and dread about what lies ahead. Despite these immense challenges, the study found that Gazans have developed various coping mechanisms to endure their circumstances. Social support from family and community, religious faith and a sense of humor emerged as key strategies for resilience. These coping methods provide emotional and practical assistance, helping individuals navigate the extreme adversity they face daily.

The findings of this study highlight the profound humanitarian crisis in Gaza, where the loss of homes, loved ones and basic human dignity have created an environment of extreme precarity. Yet, the resilience displayed through social bonds, faith and humor reflects the enduring human spirit in the face of overwhelming adversity. This study calls for urgent attention to the ongoing crisis in Gaza, advocating for comprehensive humanitarian support and psychological interventions to address the deep-seated trauma and help rebuild lives shattered by conflict.

As researchers from Palestine, we recognize that our cultural heritage and Palestinian identity influence our approach to this study. Having grown up in Occupied Palestine, we have been shaped by the challenging realities of political conflict and ongoing trauma. In conducting this research, we are mindful that our positionality may affect how we interpret data and interact with participants. We are committed to maintaining a reflective stance to ensure transparency and awareness of these influences throughout the research process.

**Open peer review.** To view the open peer review materials for this article, please visit http://doi.org/10.1017/gmh.2025.23.

**Data availability statement.** The datasets used and/or analyzed during the current study are available from the corresponding author on reasonable request.

**Author contribution.** All authors contributed equally to this article.

**Financial support.** No funding was received for this study.

**Competing interest.** The authors declare that they have no competing interests. All authors agreed in submitting the manuscript to the journal.

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
