## [Reviewer Report]

This paper is exceptionally well-executed and of immense interest. The scientific literature greatly benefits from publications that provide insight into the situation in Gaza following these 11 months of what can be described as genocide. The work is written in a clear, concise, and thoroughly structured manner. The introduction is comprehensive, up-to-date, and deeply explores the complex connections surrounding colonial trauma. The interviews presented are highly compelling, and I eagerly anticipate the publication of this work so that I can reference it in my own research. I commend the authors for undertaking such a challenging project.

There are, however, two minor suggestions for improvement. First, it might be valuable to include a brief paragraph regarding the authors' positionality. Are they Palestinian, or are they from other countries with a focus on Palestine? This information would be beneficial to the reader. Lastly, two very small issues: please review the punctuation and formatting (occasionally there are capitalizations where they shouldn’t be, or inconsistent line spacing), and on page 6, line 36, there is a missing reference.

---

## [Reviewer Report]

This paper is a well-written and methodologically sound contribution to the scientific literature. The insights it provides into the situation in Gaza are particularly valuable, especially given the context of recent and ongoing conflicts. The work is clear, concise, and well-structured overall. The introduction offers a comprehensive and current exploration of the complex connections surrounding colonial trauma. However, due to the density of the material, the introduction could benefit from the addition of one or two paragraphs with subheadings to help guide the reader more effectively.

I suggest renaming the initial section from “Theoretical Background” to “Introduction.” The current section does not present compelling theories that directly lead the research; instead, it provides a foundational overview that better fits the role of an introduction.

The interviews included in the study are compelling and add significant depth to the research. However, the discussion section could be improved by emphasizing the novelty of the work more prominently at the beginning. Highlighting the uniqueness of this research, especially considering it was conducted during an active conflict, will underscore its contribution to the field. It is crucial to clearly articulate how this study stands out and the new insights it provides in the context of ongoing conflict.

The value of this work is substantial, particularly in its timely and important examination of the situation in Gaza. The study is worthy of publication not only for its contribution to the academic discourse but also for its role in promoting Palestinian research and solidarity during such a challenging period. Emphasizing these aspects will enhance the recognition of the research’s significance and its impact on the field.

Additionally, the discussion and conclusion sections would benefit from incorporating more literature to support their statements and interpretations. Specifically, many articles—both empirical and theoretical—have been published regarding the suffering of Gaza’s population after October 7. Including references to these sources will strengthen the arguments and provide a more robust context for the findings.

More information about the sample selection process would also be helpful. Understanding how participants were chosen, especially in such a challenging environment, is crucial for assessing the study’s validity. Given the difficult circumstances surrounding data collection in Gaza, it would be beneficial for the authors to discuss how these challenges were addressed and managed during the research process.

While the description of the findings is concise, it could be expanded to provide additional context and analysis. In particular, incorporating research on the consequences of the events of October 7 on the Palestinian population would add further depth and relevance to the study.

In terms of ethics, the paper would benefit from a more detailed reflection on the principle of “do no harm.” This should include considerations of how the research design and implementation have safeguarded participants' well-being and minimized potential risks, given the sensitive nature of the topic and the context in which the study was conducted.

I also have a few minor suggestions:

It would be helpful to include a brief paragraph on the authors' positionality. Clarifying whether the authors are Palestinian or come from other countries with a focus on Palestine would provide important context for the reader.

Please review the punctuation and formatting, as there are instances of unnecessary capitalizations and inconsistent line spacing.

Additionally, I recommend checking the consistency between references cited in the text and those listed in the reference section, as it seems that some references might have been missed.

Finally, I suggest using the terms “displaced” or “internally displaced people” instead of “refugee” where applicable, as these terms may more accurately describe the population discussed in the study.

Overall, these are mild suggestions meant to enhance the clarity and impact of an already valuable contribution. The paper’s worthiness for publication, especially in such difficult times, underscores its importance in advancing Palestinian research and fostering solidarity.

---

## [Reviewer Report]

Great article and very relevant in this context.

Please review the keywords again.

Please ensure that the references are APA.

---

## [Editor Report]

Please pay attention to all three reviewers' comments and kindly improve the manuscript, considering their input.

---

## [Reviewer Report]

The authors have addressed all the points raised in the previous review comprehensively and effectively. This paper is a well-written and methodologically sound contribution to the scientific literature. Its insights into the situation in Gaza are particularly valuable, especially given the context of recent and ongoing conflicts.

The introduction has been improved with the addition of subheadings, which enhance the structure and guide the reader more effectively. Renaming the initial section to “Introduction” was a good decision, as it now aligns better with the content’s purpose.

The discussion section now emphasizes the novelty of the work and its unique contribution, particularly given the challenges of conducting research during an active conflict. The inclusion of recent literature and references related to the events of October 7 has strengthened the analysis and contextualized the findings more robustly.

Additionally, the authors have clarified the sample selection process, addressed challenges related to data collection, and provided an expanded discussion of ethical considerations, including the principle of “do no harm.” The inclusion of a paragraph on the authors' positionality adds transparency and further contextualizes the study.

The formatting and reference inconsistencies have been resolved, and the terminology has been refined, ensuring greater accuracy and precision throughout the paper.

Overall, this study is a timely and significant contribution, advancing academic discourse and promoting Palestinian research during a critical period. I am pleased to recommend this work for publication.